# Method for Fault Diagnosis of Temperature-Related MEMS Inertial Sensors by Combining Hilbert–Huang Transform and Deep Learning

**DOI:** 10.3390/s20195633

**Published:** 2020-10-01

**Authors:** Tong Gao, Wei Sheng, Mingliang Zhou, Bin Fang, Futing Luo, Jiajun Li

**Affiliations:** 1School of Instrumentation Science and Opto-electronics Engineering, Beihang University, 37, Xueyuan Road, Haidian District, Beijing 100083, China; by1517120@buaa.edu.cn; 2State Key Laboratory of Internet of Things for Smart City Faculty of Science and Technology, University of Macau, Macau SAR 999078, China; mingliangzhou@cqu.edu.cn; 3School of Computer Science, Chongqing University, 174 Shazheng Street, Shapingba District, Chongqing 400044, China; fb@cqu.edu.cn (B.F.); lft@cqu.edu.cn (F.L.); lijiajun@cqu.edu.cn (J.L.)

**Keywords:** fault diagnosis, Hilbert-Huang transform, BLSTM, CNN

## Abstract

In this paper, we propose a novel method for fault diagnosis in micro-electromechanical system (MEMS) inertial sensors using a bidirectional long short-term memory (BLSTM)-based Hilbert–Huang transform (HHT) and a convolutional neural network (CNN). First, the method for fault diagnosis of inertial sensors is formulated into an HHT-based deep learning problem. Second, we present a new BLSTM-based empirical mode decomposition (EMD) method for converting one-dimensional inertial data into two-dimensional Hilbert spectra. Finally, a CNN is used to perform fault classification tasks that use time–frequency HHT spectrums as input. According to our experimental results, significantly improved performance can be achieved, on average, for the proposed BLSTM-based EMD algorithm in terms of EMD computational efficiency compared with state-of-the-art algorithms. In addition, the proposed fault diagnosis method achieves high accuracy in fault classification.

## 1. Introduction

The use of unmanned aerial vehicles (UAVs) is growing due to their advantages in mobility and economics. The reliability achieved by sensors as a measurement and control system has a great impact on the performance of UAVs [1,2,3]. Inertial sensors in UAVs measure the states of motion. Because micro-electromechanical system (MEMS) inertial sensors have obvious advantages in weight, cost, and power consumption, MEMS inertial sensors are widely used in UAVs to perform inertial measuring tasks. However, the performance of MEMS inertial sensors can be significantly affected by external temperature [4,5,6] so diagnosing this type of fault is crucial in guaranteeing the reliable control of UAVs.

Fault diagnosis (FD) performs tasks such as fault detection [7,8,9] and fault tolerance [10,11] There are three main types of FD methods: hardware-redundant, model-based, and data-driven methods. Hardware-redundant FD handles the faults using redundant devices [12,13,14]. which brings additional costs and weight consumption. Model-based FD adopts analytical models [15,16,17,18,19] to pattern the faults and relies on the precision and accuracy of mathematical models. Data-driven FD recognizes the fault according to the fault features found from a large volume of historical data [20,21,22]. This method has low modeling dependency and has attracted much interest from researchers [23,24,25].

Data-driven methods have achieved high performance in aircraft FD applications and have focused on aircraft actuator FD [26], aircraft sensors FD [27], aero engine systems FD [28,29], and fault data analysis [30,31]. Deep learning (DL) learns features from big data [32,33] and avoids the complex processes stemming from handcrafted features. CNN is a powerful DL model for handling two-dimensional (2-D) images and has been used in FD research, such as mechanical systems FD [34,35], circuit systems FD [36], and avionics FD [37]. In FD applications, because raw data is often sampled in one-dimensional (1-D) format, researchers have turned to feature extraction operations that construct 2-D features for addressing FD problems using CNNs, such as sliding window [38,39], short time Fourier transform (STFT) [40], discrete wavelet transform (DWT) [41,42], and Hilbert–Huang transform (HHT) [43,44]. Structure health monitoring (SHM) is becoming a research hotspot in which CNN is applied and several methods have been proposed in the field of SHM combining CNN to solve mechanical system SHM problems [45,46,47].

HHT is a time–frequency analysis method that consists of empirical mode decomposition (EMD) and Hilbert transform (HT), and it offers high performance in analyzing nonstationary signals by adopting EMD to compute intrinsic mode functions (IMFs). Many researchers have adopted HHT-based methods to perform data analysis tasks [48,49,50]. The FD methods combining HHT and CNN are becoming research hotspots in different applications, such as power distribution system FD [51], geared system FD [52] and pipeline system FD [53]. Our research aims to recognize fault patterns hidden in large volume of real MEMS inertial sensors measurement data that are non-stationary due to the affection of variable temperature condition, study on HHT-based feature extraction algorithm offers a reliable fault feature representation strategy on handling nonstationary MEMS temperature related faults.

LSTM [54] is a type of recurrent neural network (RNN) used in serial information processes. As a DL model for processing sequential information, LSTM can directly process the signals of FD, which is the reason of the frequently usage of LSTM in FD study, such as useful life prediction [55], turbine FD [56], and gearbox FD [57].

Analysis on large volume of fault data of our MEMS inertial sensors shows that the temperature related MEMS faults have obvious non-linear and non-stationary characters, DL-based methods offer an effective way to recognize these faults in end-to-end manner. Although the above methods focus on applying deep learning methods to FD, these methods still have the following shortcomings for the UAV MEMS inertial sensors FD:

First, there is little research focused on adopting DL-based FD methods to solve the problems of MEMS inertial sensor FD. Second, traditional 1-D to 2-D data converting methods have limitations in handling nonstationary and nonlinear MEMS measurements. Sliding window-based methods can hardly acquire temperature-related fault features without a full range of temperature variation. Fourier transform-based methods such as STFT and DWT have limitations on processing nonstationary signals. Third, HHT has advantages in processing nonlinear and nonstationary signal, but current HHT-based methods need much iterative computation caused by the two-layer loops iteration method, which decreases computing efficiency. Fourth, LSTM is a powerful tool for performing sequential related tasks, and it can introduce the ability of using prior knowledge to HHT, which is useful in increasing HHT efficiency. However, there are no studies focusing on this strategy.

To address these problems, we have proposed a MEMS inertial sensor fault diagnosis method by combining BLSTM-based HHT and CNN. The contributions of our work are as follows:(1)We formulate the MEMS inertial sensor fault diagnosis method for deep learning problems. The proposed FD method uses the BLSTM-based HHT method to perform the feature extracting task and adopts a CNN to classify MEMS inertial sensor faults.(2)We propose a BLSTM-based HHT algorithm that performs a direct IMF computation method by introducing BLSTM to EMD, thereby improving the EMD efficiency by decreasing the number of iterations in obtaining IMFs. Additionally, we adopt noise assistance and frequent shifting to further improve the HHT performance.(3)We use the multiscale CNN (MS-CNN) to perform the fault classification task, in which Hilbert spectrums generated by the proposed BLSTM-based HHT are used as the input of the MS-CNN model.

The remainder of this paper is constructed as follows: Section 2 reviews the related works on the proposed FD method. Section 3 describes the proposed FD method. Section 4 presents the detailed experiments. Section 5 presents the authors’ conclusions.

## 2. Related Works

### 2.1. CNN-Based Fault Diagnosis

CNNs have various applications in image processing and are used in data-driven FD methods for complex systems. In [20], Guo et al. proposed a method for FD of UAV sensors by jointing an extended Kalman filter, STFT, and CNN. This FD method adopted CNNs to recognize various fault features, and the STFT is used to convert the 1-D signals to 2-D spectrums. In [34], Yao et al. presented a multi-scale CNN-based gear FD method. This method designs an attention mechanism based on multi-scale CNN to mine the relevant fault information and uses multi-scale CNN to recognize the faults. In [35], a hierarchical CNN-based FD method was proposed by Guo et al. for performing a mechanical FD task. In [36], Wang et al. proposed a 1-D CNN (1D-CNN)-based Hiller system fault diagnosis method that combines 1D-CNN and a gated recurrent unit to perform the fault identification task. In [37], a random oversampling-based CNN FD method was presented by Chen et al. to handle fault confusing problems and is applied in avionics FD. In [38], Wen et al. presented a CNN-based FD algorithm in which a CNN is used to learn fault features from reconstructed raw data directly without complex feature extracting operation. In [41], Aziz et al. adopted a 2-D CNN to perform the task of fault detection in photovoltaic (PV) arrays, with the CNN used to extract fault features in PV system scalograms.

SHM monitors the failure or degeneration of components in complex systems and plays an important role in maintenance and activation inspection, researcher have used CNN to perform the complex SHM tasks. In [45], Tabian et al. proposed a CNN-based meta-model to address the impact monitoring problem of complex composite structures, the method transferred the piezoelectric sensors signal to 2D images and used CNN to perform the health state classifications, this method has advantages in effective end-to-end state monitoring and well transferability to complex structures. In [46], Oliveira et al. proposed a novel electromechanical impedance SHM solution by combining electromechanical impedance piezoelectricity (EMI-PZT), the high accuracy of proposed method is guaranteed by CNN-based feature extraction which include several banks of filters. In [47], a graphic model and CNN-based bolt loosening SHM method was proposed by Pham et al. The proposed method performed a CNN-based bolt detection algorithm and used the Hough transfer-based method to estimate the bolt angle, this method proposed a convenient strategy to monitoring the bolt structures just using images sampled by a camera.

### 2.2. HHT-Based Fault Diagnosis

HHT is a new time–frequency method based on empirical mode decomposition (EMD) and Hilbert transform (HT), that has advantages in decomposing the frequency components of nonlinear and nonstationary signals.

Researchers are using HHT to perform signal analysis. In [48], Bagherzadeh et al. enhanced the EMD algorithm using multi-object optimization, which introduced a genetic algorithm to determine the best decision parameters. The proposed method is used to detect ice formation on aircraft. In [49] Zheng et al. presented a flutter test method using HHT, and the method mitigated the mode mixing effect in HHT. In [50], an improved EMD, called IEEMD, added Gaussian noise into the EMD operation. The method, presented by Mokhtari et al., was used to identify the dynamic models of aircrafts.

As a time–frequency-based data converting method, the output of HHT can include the features to train the models in data-driven FD. In [44], Sheng et al. proposed a fault location method for power systems that employed the high frequency components of EMD to train CNNs to learn the features of the faults. In [58], a modified HHT constructed time- and frequency-domain–based nonlinear entropy features to reduce the effects of noise. Researchers have studied FD methods that combine HHT-based feature extraction and deep learning-based FD methods. In [43], Yang et al. proposed an opening damper evaluation method, which used HHT to convert a 1-D signal into 3-D time-frequency images and used these images for training CNNs in order to evaluate the vibration of power transmission systems. In [59], Liang et al. used the intrinsic mode function (IMF) components of EMD to construct the fault features, which were then used to train the CNN-based FD network, which was called CRNN. In [60], Chen et al. proposed an EMD-based decomposing method, called adaptive sparsest narrow-band decomposition (ASNBD) and applied the ASNBD to FD in roller bearings. In [51], a power distribution Fault classification method combining HHT and CNN is proposed by Guo et al., the proposed method adopted HHT to construct the time-frequency energy map and used CNN to classify the faults patterns. In [52], Han et al. employed a HHT-CNN- based method to address the geared system FD problem, the presented FD method used the two-dimension HHT spectrum of vibration acceleration signals as the input of CNN and the trained CNN is used to perform the fault classification task. In [53], Xie et al. used the HHT and CNN to address the pipeline leakage detection problem, in this paper, acoustic signals are converted to time-frequency image waves by HHT and the converted images are fed to a two-layer CNN for performing the leakage detecting task.

### 2.3. LSTM-Based Fault Diagnosis

LSTM is a well-known RNN model that can process data combined with contextual information and has been successfully applied in serial information processing. LSTM also has been used in FD-related research. In [61], Huang et al. adopted a BLSTM-based method to solve the prognostic problem of aircraft engine remaining useful life (RUL). In [62], Yang et al. proposed a method for FD for aircraft electromechanical actuators that used LSTM to analyze the correlation of sensors. In [63], an aircraft engine degradation assessment and RUL prediction framework based on LSTM was proposed by Miao et al.

## 3. Method for Fault Diagnosis of MEMS Inertial Sensors in Unmanned Aerial Vehicles by Combining Hilbert–Huang Transform and Deep Learning

### 3.1. Overview

The proposed FD method performs an end-to-end FD strategy that uses the inertial data (from the measurements of a gyroscope or accelerometer) as input and outputs the fault classifications. The proposed FD method includes two operations: feature extraction and fault classification. The feature extraction task converts the one-dimension inertial measurement data to a time-frequency spectrum by the proposed BLSTM-based HHT. The fault classification task classifies the inertial sensors’ fault states by CNN.

The proposed FD method is shown in Figure 1. First, the inertial data is processed by a frequency shifting operation. Second, a BLSTM-based EMD algorithm is performed to obtain the IMFs, and the noise assistance algorithm is used to process the signals of computing different IMFs. Third, we use Hilbert transformation to construct the Hilbert spectrum, which is used to feed the 2-D features to the CNN. Finally, the MS-CNN outputs the fault classifications.

### 3.2. Proposed BLSTM-Based HHT

In this section, we formulate the proposed HHT method. The proposed HHT structure is shown in Figure 2. First, the frequency shifting [49] is applied to reduce the mode mixing. Second, the BLSTM-based EMD is performed to compute the IMFs of inertial data and noise assistance analysis. Ensemble empirical mode decomposition (EEMD) [1] is used to further reduce the mode mixing. Finally, the Hilbert-transform (HT) is performed so that the IMFs cam obtain the time–frequency spectrum.

#### 3.2.1. Modeling of End-to-End EMD

In traditional EMD, IMFs are obtained by two iteration loops. The inner loop computes each IMF by trend signal iteration, while the outer loop removes the IMF from the raw signal for the purpose of computing the next IMF. The inner loop is iterated as:(1)$$h_{i,1}\left(t\right)=r_{i}\left(t\right)$$
(2)$$h_{i,2}\left(t\right)=h_{i,1}\left(t\right)-m_{i,1}\left(t\right)$$
(3)$$h_{i,j+1}\left(t\right)=h_{i,j}\left(t\right)-m_{i,j}\left(t\right)$$
where $h_{i,j}\left(t\right)$ indicates the weakened component of the *j-*th iteration for computing the *i*th IMF, $m_{i,j}\left(t\right)$ is the trend signal in each iteration, which is the average of the upper and lower envelop curve of weakened residual $h_{i,j-1}\left(t\right)$, as follows:(4)$$m_{i,j}\left(t\right)=\frac{u_{i,j}\left(t\right)-l_{i,j}\left(t\right)}{2}$$
where $u_{i,j}\left(t\right)$ and $l_{i,j}\left(t\right)$ are the upper and lower values of the envelop curve of $h_{i,j}\left(t\right)$ by crossing the cubic splines.

Once the trend signal is removed from weakened component $h_{i,j}\left(t\right)$, the *i-*th IMF can be obtained. Then the inner iteration is stopped and the final weakened component is deemed as the *i-*th IMF. Then EMD removes the *i-*th IMF from the residual $r_{i}\left(t\right)$ and obtains the new residual, as follows:(5)$$r_{i+1}\left(t\right)=r_{i}\left(t\right)-IMF_{i}$$
where $IMF_{i}$ indicates the *i-*th IMF.

During the EMD iterations, the trend signal $m_{i,j}\left(t\right)$ is the residual containing the nonlinear and the non-zero average components. We can use polynomial fitting to indicate the trend signal. Assume that the best weakened component is obtained in the *J-*th iteration, the final iteration is:(6)$$IMF_{i}=h_{i,J}\left(t\right)=h_{i,J-1}\left(t\right)-m_{i,J-1}\left(t\right)$$

The (*J*−1)-th weakened component can be written by the (*J*−2)-th weakened component as:(7)$$h_{i,J-1}\left(t\right)=h_{i,J-2}\left(t\right)-m_{i,J-2}\left(t\right)$$

Thus, the $IMF_{i}$ can be expressed as the (*J*−2)-th weakened component and the sum of the (*J*−2)-th and the (*J*−1)-th trend signals:(8)$$\begin{array}{ll} IMF_{i} & =h_{i,J-1}\left(t\right)-m_{i,J-1}\left(t\right)\\ & =h_{i,J-2}\left(t\right)-m_{i,J-2}\left(t\right)-m_{i,J-1}\left(t\right) \end{array}$$

Furthermore, $IMF_{i}$ can be deemed as the sum of the initial weakened component $h_{i,1}\left(t\right)=r_{i}\left(t\right)$ and the negative total trend signal that contains all trend signals, from the first to the last:(9)$$\begin{array}{ll} IMF_{i} & =h_{i,1}\left(t\right)\\ & \;-\left(m_{i,1}\left(t\right)+m_{i,2}\left(t\right)+\ldots +m_{i,j}\left(t\right)+\ldots +m_{i,J-1}\left(t\right)\right)\\ & =r_{i}\left(t\right)-\sum_{j=1}^{J-1}m_{i,j}\left(t\right)\\ & =r_{i}\left(t\right)-M_{i}\left(t\right) \end{array}$$
where $M_{i}\left(t\right)$ indicates the sum of all trend signals of the iterations. It is worth noting that this iteration is necessary for irregular signals; we cannot obtain the $M_{i}\left(t\right)$ directly according to $r_{i}\left(t\right)$ without specific abstract features. However, there are specific abstract features in the inertial data, which makes directly obtaining $M_{i}\left(t\right)$ from $r_{i}\left(t\right)$ feasible by training an end-to-end regression model, as follows:(10)$$\left\{ \begin{array}{l} M_{i}\left(t\right)=f\left(r_{i}\left(t\right),\Theta \right)\\ M_{i}\left(t\right)=r_{i}\left(t\right)-IMF_{i} \end{array}\right.$$
where Θ indicates the parameters of the regression model.

Obviously, this method decreases the iteration epochs for computing IMF. In this regression model, $M_{i}\left(t\right)$, which is obtained according to $r_{i}\left(t\right)$ and the $IMF_{i}$ by inner loop sifting, can be treated as the training label and the $r_{i}\left(t\right)$ is the input feature. The $IMF_{i}$ can also be obtained by the *j-*th weakened component and trend signals from the (*j*+1)-th to the end. Thus, Equations (9) and (10) could be expressed by Equations (11) and (12), as follows:(11)$$\begin{array}{ll} IMF_{i} & =h_{i,j}\left(t\right)-\left(m_{i,j+1}\left(t\right)+\ldots +m_{i,J-1}\left(t\right)\right)\\ & =h_{i,j}\left(t\right)-\sum_{\beta =j+1}^{J-1}m_{i,\beta }\left(t\right)\\ & =h_{i,j}\left(t\right)-M_{i,j}\left(t\right) \end{array}$$
(12)$$\left\{ \begin{array}{l} M_{i,j}\left(t\right)=f\left(h_{i,j}\left(t\right),\Theta \right)\\ M_{i,j}\left(t\right)=h_{i,j}\left(t\right)-IMF_{i} \end{array}\right.$$

#### 3.2.2. BLSTM-Based Sifting

We formulate the traditional sifting process of EMD into an end-to-end regression task. In this research, we propose a BLSTM-based coefficient regression algorithm to obtain the trend signal $M_{i}\left(t\right)$ from $r_{i}\left(t\right)$ directly.

Equation (10) represents an efficient way to obtain the training data set, in that the regression label, trend signal $M_{i}\left(t\right)$, can be obtained by removing ${IMF}_{i}$ from $r_{i}\left(t\right)$.

$M_{i}\left(t\right)$ can be represented by piecewised segments, as follows:(13)$$M_{i}^{}\left(t\right)=\sum_{k=1}^{K-1}M_{i}^{H}\left(k\right)$$
where k∈{1,….,K−1} indicates the number of segments of trend signal $M_{i}^{}\left(t\right)$, which is divided by local extrema points of $r_{i}\left(t\right)$, and $M_{i,k}^{H}\left(t\right)$ indicates the piecewised trend signal that is computed using cubic polynomial fitting as follows:(14)$$M_{i}^{H}\left(k\right)=\left\{ \begin{array}{l} a_{i,k}^{H}t^{3}+b_{i,k}^{H}t^{2}+c_{i,k}^{H}t+d_{i,k}^{H},t_{k-1}\leq t<t_{k}\\ 0,otherwise \end{array}\right.$$
where $\left(a_{i,k}^{H},b_{i,k}^{H},c_{i,k}^{H},d_{i,k}^{H}\right)$ indicate the coefficients of fitting in each segment. The local extrema points are generated from $r_{i}\left(t\right)$, and this makes each segment in $r_{i}\left(t\right)$ monotonic, so that it can be easily fitted by a cubic polynomial. Furthermore, the trend signal is the residual by removing the high-frequency ${IMF}_{i}$ from $r_{i}\left(t\right)$, which means that piecewised fitting can aptly describe a “low frequency” trend signal. Note that Equation (14) presents a strategy to code data in each segment to a fixed-dimension expression, which is essential for BLSTM.

Thus, according to Equations (10), (13), (14), the regression task is to recognize the piecewise cubic fitting parameters $\left(a_{i,k}^{H},b_{i,k}^{H},c_{i,k}^{H},d_{i,k}^{H}\right)$ from the piecewised $r_{i}\left(t\right)$. We use BLSTM to address this regression problem, which is described as:(15)$$\left(a_{i,k}^{H},b_{i,k}^{H},c_{i,k}^{H},d_{i,k}^{H}\right)=BLSTM\left(r_{i}\left(k\right),\Theta \right),k\in \left\{1,2,\ldots ,K-1\right\}$$
where $r_{i}\left(k\right)$ indicates the piecewised $r_{i}\left(t\right)$ by local extrema points, k indicates the segment index, and K is the amount of local extrema points.

Different IMFs have different scales, which lead to reduced performance in the convergence and learning efficiency in the case of feeding unnormalized data into the neural network directly. Thus, we adopt the min-max normalization method to normalize the weakened component data, and the normalization and piecewised operation are:(16)$$\left\{ \begin{array}{l} \hat{r_{i}}\left(t\right)=\frac{r_{i}\left(t\right)-\min\left(r_{i}\left(t\right)\right)}{\max\left(r_{i}\left(t\right)\right)-\min\left(r_{i}\left(t\right)\right)}\\ \hat{r_{i}}\left(k\right)=pieceswise\left(\hat{r_{i}}\left(t\right)\right),k\in \left\{1,2,\ldots ,K-1\right\} \end{array}\right.$$

The length of each piecewise weakened component is different due to the nonstationary and the non-linear nature of the raw signal. To meet the fixed input dimension length requirement of BLSTM, we first perform a cubic polynomial fitting on $\hat{r_{i}}\left(k\right)$, then the coefficient of $\hat{r_{i}}\left(k\right)$ is used as the input of BLSTM. The operation is as follows:(17)$$\left\{ \begin{array}{l} \hat{r_{i}}\left(k\right)=\left\{ \begin{array}{l} p_{i,k,1}t^{3}+p_{i,k,2}t^{2}+p_{i,k,3}t+p_{i,k,4},t_{k-1}\leq t<t_{k}\\ 0,otherwise \end{array}\right.\\ \begin{array}{ll} p_{i}\left(k\right) & =\left[p_{i,k,1},p_{i,k,2},p_{i,k,3},p_{i,k,4}\right]\\ & =cubic\;polynomial\;fitting\left(\hat{r_{i}}\left(k\right)\right) \end{array} \end{array}\right.$$
where $\hat{r_{i}}\left(k\right)$ indicates the normalized segment obtained by local extrema points, and $p_{i}\left(k\right)$ indicates the polynomial fitting coefficient.

Thus, the BLSTM model in Equation (15) can be presented as:(18)$$\begin{array}{ll} p_{i}^{H}\left(k\right) & =\left(a_{i,k}^{H},b_{i,k}^{H},c_{i,k}^{H},d_{i,k}^{H}\right)\\ & =BLSTM\left(p_{i}\left(k\right),\Theta \right),k\in \left\{1,2,\ldots ,K-1\right\} \end{array}$$

Additionally, the template conditions of inner loop sifting can augment the robustness of the recognition pattern, and the weakened component-based trend signal regression model can be presented as:(19)$$\left\{ \begin{array}{l} {\hat{h}}_{i,j}\left(k\right)=\left\{ \begin{array}{l} q_{i,j,k,1}t^{3}+q_{i,j,k,2}t^{2}+q_{i,j,k,3}t+q_{i,j,k,4},t_{k-1}\leq t<t_{k}\\ 0,otherwise \end{array}\right.\\ \begin{array}{ll} q_{i,j}\left(k\right) & =\left[q_{i,j,k,1},q_{i,j,k,2},q_{i,j,k,3},q_{i,j,k,4}\right]\\ & =cubic\;polynomial\;fitting\left({\hat{h}}_{i,j}\left(k\right)\right) \end{array} \end{array}\right.$$
(20)$$\begin{array}{ll} q_{i,j}^{H}\left(k\right) & =\left(a_{i,j,k}^{T},b_{i,j,k}^{T},c_{i,j,k}^{T},d_{i,j,k}^{T}\right)\\ & =BLSTM\left(q_{i,j}\left(k\right),\Theta \right),k\in \left\{1,2,\ldots ,K-1\right\} \end{array}$$
where the annotation i,j indicates the *j*th iteration for computing *i*th *IMF*, ${\hat{h}}_{i,j}\left(k\right)$ indicates the normalized weakened component, $q_{i,j}^{H}\left(k\right)$ indicates the cubic polynomial coefficients of ${\hat{h}}_{i,j}\left(k\right)$, k indicates the *k*th segment, and $q_{i,j}^{H}\left(k\right)$ indicates the output of BLSTM model with input $q_{i,j}\left(k\right)$.

The structure of the proposed BLSTM-based sifting method is shown in Figure 3. The proposed BLSTM-based sifting algorithm is based on a deep BLSTM regression model that consists of several BLSTM layers and one output layer. The neurons in the BLSTM layer can be modified to mine detailed features. The output layer consists of a three-layer fully connected network with 32 neurons in the first layer, 16 neurons in the second layer, and four neurons in the third layer. The output layer is used to reconstruct the features from the BLSTM layers to four dimensions, which is equal to the dimensions of polynomial parameters of the trend signal.

The input of the BLSTM model can be formulated as:(21)$$\begin{array}{l} \vec{X}=\overset{\leftarrow }{X}=\left[\vec{x}\left(1\right),\vec{x}\left(2\right),\ldots ,\vec{x}\left(k\right),\ldots \vec{x}\left(K-1\right)\right]\\ =\left[\overset{\leftarrow }{x}\left(1\right),\overset{\leftarrow }{x}\left(2\right),\ldots ,\overset{\leftarrow }{x}\left(k\right),\ldots \overset{\leftarrow }{x}\left(K-1\right)\right]\\ =\left[p_{i}\left(1\right),p_{i}\left(2\right),\ldots ,p_{i}\left(k\right),\ldots ,p_{i}\left(K-1\right)\right] \end{array}$$
where $\vec{x}\left(k\right)$ and $\overset{\leftarrow }{x}\left(k\right)$ are the inputs of the forward procedure and the backward procedure and K−1 indicates the total number.

The regression task of the BLSTM is formulated as follows:(22)$$\begin{array}{l} \overset{↔}{H}=\left[\overset{↔}{h}\left(1\right),\overset{↔}{h}\left(2\right),\ldots ,\overset{↔}{h}\left(k\right),\ldots ,\overset{↔}{h}\left(K\right)\right]\\ =f\left(\vec{X};{\vec{\Theta }}_{LSTM}\right)⊕f\left(\overset{\leftarrow }{X};{\overset{\leftarrow }{\Theta }}_{LSTM}\right) \end{array}$$
where $\overset{↔}{h}\left(k\right)$ indicates the output of BLSTM, which is the coefficient of piecewise trend signal:(23)$$\overset{↔}{h}\left(k\right)=\left(a_{i,k}^{H},b_{i,k}^{H},c_{i,k}^{H},d_{i,k}^{H}\right)$$

#### 3.2.3. BLSTM Training Strategy

The performance of BLSTM-based EMD is dependent on the training dataset. We propose a strategy to generate the dataset, which is a modified EMD operation that records data and combines various efficient sifting methods to obtain the optimal IMFs.

Figure 4 indicates the modified EMD with data recording operation. Note that the symbol “^” indicates the normalized signal. The proposed three-part method includes the proposed EMD, feature coding, and feature recording. The proposed EMD consists of frequency shifting, classic EMD, and noise assistance analysis. The first step is the frequency shifting operation [49] The second step is normalization, as seen in Equation (16), to obtain the normalized residual ${\hat{r}}_{i}\left(t\right)$; then the processed data is fed to the classic EMD framework. Note that the classic EMD in Figure 4 is represented as an inner loop and outer loop. The inner loop is used to compute each IMF and the outer loop is used to compute the IMFs. The outer loop of the proposed EMD adopts the EEMD [1] to reduce the mode mixing. The feature coding operation converts the data of the improved EMD to the four-dimension fitting parameters. The first operation of the feature coding is piece wising the residual ${\hat{r}}_{i}\left(t\right)$. Each ${\hat{r}}_{i}\left(t\right)$ in an EMD operation is piecewised by the extrema points of the original signal $\hat{x}\left(t\right)={\hat{r}}_{1}\left(t\right)$, as follows:(24)$$\left\{Ex\right\}=\left\{ex_{1}\left(1\right),ex_{1}\left(2\right),\ldots ,ex_{1}\left(k\right),\ldots ,ex_{1}\left(K-1\right)\right\}$$
where $ex_{1}\left(k\right)$ indicates the time stamp of each local extrema of ${\hat{r}}_{1}\left(t\right)$. EMD is a method that removes frequency components from residual, which means that the local extrema points {Ex} that partition a high-frequency signal into (K−1) monotonical intervals can partition low-frequency weakened components without any information missing. After the {Ex(i)} is obtained, the weakened component ${\hat{h}}_{i,j}\left(t\right)$, temporary trend ${\hat{M}}_{i,j}\left(t\right)={\hat{h}}_{i,j}\left(t\right)-{\hat{IMF}}_{i}$, and trend ${\hat{M}}_{i}\left(t\right)={\hat{r}}_{i}\left(t\right)-{\hat{IMF}}_{i}$ are piecewised according to {Ex}. The final step of feature coding is cubic polynomial fitting each segment of these piecewised signals, and this is what generates the polynomial coefficients. The data recording operation creates the dataset for training the BLSTM model.

The basic dataset consists of the residual polynomial coefficients and the trend coefficients. The residual polynomial coefficients, $p_{i}\left(k\right)=\left\{p_{i,k,1},p_{i,k,2},p_{i,k,3},p_{i,k,4}\right\}$, are the training features. The trend coefficients, $p_{i}^{H}\left(k\right)=\left\{a_{i,k}^{H},b_{i,k}^{H},c_{i,k}^{H},d_{i,k}^{H}\right\}$, are the regression labels. The extension dataset consists of the “weakened component polynomial coefficients” and the “temporary trend coefficients”. The weakened component polynomial coefficients, $q_{i,j}\left(k\right)=\left\{q_{i,j,k,1},q_{i,j,k,2},q_{i,j,k,3},q_{i,j,k,4}\right\}$, are the training features. The temporary trend coefficients, $q_{i,j}^{H}\left(k\right)=\left\{a_{i,j,k}^{T},b_{i,j,k}^{T},c_{i,j,k}^{T},d_{i,j,k}^{T}\right\}$, are the regression labels. The extension dataset contains the temporary details of EMD, and it can be used to improve the robustness of BLSTM.

#### 3.2.4. IMF Computation

After the BLSTM model is trained using the basic dataset and the extension dataset, a direct EMD, as in Equation (10), can be driven by the BLSTM, as in Algorithm 1. The BLSTM-based EMD method performs a BLSTM regression operation. First, the residual is normalized and piecewised by local extrema points. Second, the cubic parameters of each segment are computed. Third, the BLSTM model is performed to regress the trend signal. Then, the piecewise trend signal is computed by the regressed parameters. Finally, the IMF is computed by removing the recovered trend signal from the residual.

### 3.3. CNN Structure

After the 2-D spectrum is obtained by HT using IMFs, the CNN is used to learn and recognize the fault patterns. We use MS-CNN within skipping connection to perform this task. The structure of the MS-CNN is shown in Figure 5. The MS-CNN has four convolutional layers, three max-pooling layers, one multiscale layer, two fully connecting (FC) layers and one output layer. The size of the input 2-D image is 32×32. The convolution kernel size of the first three convolutional layers is 3×3, and the kernel size of the last convolutional layer is 2×2. The layer configuration is described in Table 1. A rectified linear unit (RELU) is used as the nonlinear activation function.
**Algorithm 1:** BLSTM-based EMD**Requirement:** Trained BLSTM-based EMD model by dataset in Figure 4.**Input:** Inertial measurement residual $r_{i}\left(t\right)$.**Output:**$IMF_{i}\left(t\right)$.**Begin:**1:Normalize residual: ${\hat{r}}_{i}\left(t\right)=\frac{r_{i}\left(t\right)-\min\left(r_{i}\left(t\right)\right)}{\max\left(r_{i}\left(t\right)\right)-\min\left(r_{i}\left(t\right)\right)}$;2:Piecewise: $\left\{r_{i}\left(k\right)\right\}=piecewise\left(r_{i}\left(t\right)|\left\{Ex\right\}\right)$;3:**for**k in {1,2,…,(K−1)}**do:**4: Fitting $p_{i}$:5:  $p_{i}\left(k\right)=\left\{p_{i,k,1},p_{i,k,2},p_{i,k,3},p_{i,k,4}\right\}=polynormial\left(r_{i}\left(k\right)\right)$;6:end for7:Regress trend coefficient:8: $\left\{a_{i,k}^{H},b_{i,k}^{H},c_{i,k}^{H},d_{i,k}^{H}\right\}=BLSTM\left(p_{i}\left(k\right),\Theta \right)|k\in \left\{1,2,\ldots ,\left(K-1\right)\right\}$;9:**for**k in {1,2,…,(K−1)}**do:**10: Compute piecewised trend signal:
  ${\hat{M}}_{i,k}^{H}\left(t\right)=\left\{ \begin{array}{l} a_{i,k}^{H}t^{3}+b_{i,k}^{H}t^{2}+c_{i,k}^{H}t+d_{i,k}^{H},t_{k-1}\leq t<t_{k}\\ 0,otherwise \end{array}\right.$;11:**end for**12:Compute trend signal:${\hat{M}}_{i}^{}\left(t\right)=\sum_{k=1}^{K-1}{\hat{M}}_{i,k}^{H}\left(t\right)$;13:Compute IMF:${\hat{IMF}}_{i}\left(t\right)={\hat{r}}_{i}\left(t\right)-{\hat{M}}_{i}^{}\left(t\right)$;14:Recover IMF:
 $\begin{array}{ll} IMF_{i}\left(t\right) & ={\hat{IMF}}_{i}\left(t\right)\times \left(\max\left(r_{i}\left(t\right)\right)-\min\left(r_{i}\left(t\right)\right)\right)\\ & \;+\min\left(r_{i}\left(t\right)\right) \end{array}$;15:Update $r_{i}\left(t\right)$: $r_{i}\left(t\right)=r_{i}\left(t\right)-IMF_{i}\left(t\right)$;16:**End**

### 3.4. Summary of the Proposed FD Algorithm

We summarize the proposed FD method in Algorithm 2. Note that some procedures, such as frequency shifting and noise assistance, are the same as the methods described in [50,64]. However, the emphases of the BLSTM-based EMD approaches are entirely different. First, we propose the direct trend signal estimating strategy, which avoids the sifting iterating operation and stopping judgment used in current EMD methods. Second, we introduce the BLSTM model to the task of EMD operation by using the piecewise polynomial fitting-based features coding method. This coding method offers an efficient way to feed the variation dimension data to the BLSTM model with fixed data dimension, which is essential for the BLSTM model. Third, an efficient dataset-generating method is proposed, which can automatically generate the dataset for training BLSTM without a complex data annotation operation.
**Algorithm 2:** Proposed FD method**Requirements:**  Trained BLSTM-based EMD model by dataset in Figure 4;  Trained MS-CNN by Hilbert spectrums and corresponding fault labels.**Input:** Inertial measurement residual $r_{i}\left(t\right)$.**Output:** Fault classifications.**Begin:**1:Frequency shifting:$r_{i}\left(t\right)=FrequencyShifing\left(r_{i}\left(t\right)\right)$;2:**for**i in {1,2,…,I}**do**:3: Compute $IMF_{i}$ according to Algorithm I with EEMD;4:**end for**5:Compute Hilbert spectrum by HT: $sp=HT\left(IMF_{i}\right){|}_{i\in \left\{1,2,\ldots ,I\right\}}$;6:Classify fault by MS-CNN: fault=MS−CNN(sp);7:**End.**


## 4. Experiments

In this section, we describe the experiments in which we evaluated the performance of our proposed methods and compared them with other state-of-the-art FD methods. The experiments consisted of BLSTM comparisons, EMD performance evaluations and comparisons, multiscale CNN testing, and FD comparisons. The BLSTM comparison included data recording testing and BLSTM testing, with the data recording testing performed to find the best dataset generating strategy for training BLSTM, and the BLSTM testing comparing various BLSTM structures for finding the best BLSTM model. The EMD performance evaluations of the BLSTM-based EMD involved comparisons with other EMD methods, which included decomposing performance comparison and computing efficiency performance. The multiscale CNN comparison experiment compared the hyperparameters of the CNN and selected optimal parameters. The FD comparison compared our FD method with other data-driven FD methods on our inertial dataset.

### 4.1. Setting

We selected four common temperature-related MEMS inertial sensor faults, which occurred in the production of our UAV micro guidance and navigation controllers (UAV-MGNC). The faults are shown in Figure 6. Note that the data are acquired in variable temperature conditions, the polyfit curves are used to remove the test rig-related trend term from samples, and the parting line is used to select the fault data in a temperature section that triggers the faults.

These four types of conditions from real data contain all typical states of temperature related MEMS inertial faults, precisely recognizing on these faults can address much of temperature related FD problems. For the normal condition, the measurement can be well compensated using polynomial fitting, and the residual is stationary. A fitting tendency error indicates that the residual after compensation shows the obvious trends, and to fit this error, the UAV controller should use a higher-order model to compensate the measurement, which increases the computation load of the controller. Bulge in a range of temperature is a hotspot in MEMS inertial data compensation filed [65,66,67], which is caused by the limitation of MEMS technology. Output hopping indicates a step functional characteristic in the measured data that is caused by an abnormal station in the peripheral circuit. We selected the data according to the temperature section in which the fault occurs and performed the EMD operation. The EMD results in traditional EMD are shown in Figure 7.

The MEMS inertial dataset (which consists of measurements made by gyroscopes and accelerometers) is collected from the calibration operation of our UAV-MGNCs. In the temperature calibration process, the UAV-MGNCs are fixed inside of variable temperature rig. The descriptions of UAV-MGNC and variable temperature rig are shown in Figure 8. The data are acquired in a variable temperature environment from −40 °C to 68 °C. The sample rate is set to 100 Hz and the temperature-related calibration time is set to 4 h. Of the samples, 80% are used as the training set and 20% are used as the testing set. Because the BLSTM-based HHT and CNN are two phases of the proposed FD method, the proposed BLSTM model and CNN model are trained use the same dataset. Table 2 indicates the size of the dataset and the labels.

### 4.2. BLSTM Comparison

In this section, we study the two factors that have the greatest influence on the performance of the BLSTM-based EMD method: the dataset and the BLSTM structure. The features contained in the dataset rely on the data recording strategy, and the BLSTM structure relies on the number of hidden layers and the number of neurons in each hidden layer.

We choose the hyperparameters of BLSTM, which include batch size, dropout rate, training epochs, and unfold steps. The best parameters are selected through grid research, which is performed in a deep BLSTM, as shown in Figure 3, with one BLSTM layer which has 128 neurons. The selected hyperparameters are shown in Table 3.

#### 4.2.1. Comparison of Data Recording Strategies

We test the effect of the training set with different extension datasets on BLSTM training performance. In the dataset, the feature-label pairs, $\left\{{\hat{r}}_{i}\left(k\right),\left(a_{i,k}^{H},b_{i,k}^{H},c_{i,k}^{H},d_{i,k}^{H}\right)\right\}$, are the basic BLSTM training set. The robust features are learned by using the extension dataset, $\left\{{\hat{h}}_{i,J}\left(k\right),\left(a_{i,J,k}^{T},b_{i,J,k}^{T},c_{i,J,k}^{T},d_{i,J,k}^{T}\right)\right\}$. The index J belongs to the set as follows:(25)$$J\in \left\{2,3,4,\ldots .,J_{s}\right\},J_{s}=\mathrm{int}\left(RI•\max\left(j\right)\right)$$
where max(j) indicates the maximum iteration numbers of inner loop sifting. $J_{s}$ indicates the maximum value of J, which is controlled by the parameter RI∈(0,1). That means all $\left\{{\tilde{h}}_{i,J}\left(k\right),\left(a_{i,J,k}^{H},b_{i,J,k}^{H},c_{i,J,k}^{H},d_{i,J,k}^{H}\right)\right\}$ with J in Equation (25) will be recorded to generate the extension dataset.

The recording process is shown in Table 4. We used a deep BLSTM with two BLSTM layers (one with 128 neurons and one with 192 neurons) to test the different *RI*. The deep BLSTM is trained by the basic dataset and various extension datasets. The performance of the BLSTM with different *RI*s is evaluated by training loss (the lower the training loss, the more robust the dataset). The mean square error function is the loss function. The Adam optimizer is the back-propagation algorithm. The training batch is set to 16, and there are 200 training epochs.

Figure 9 shows the training losses with different RIs. The lowest values of training loss are decreased with RI increasing. After *RI* achieves 1/5, the lowest training loss tends to be steady. That confirmed our theory that more related features of $r_{i}\left(t\right)$ enhance the robustness of BLSTM. However, the overlarge *RI* contains many features that are of no use to enhance the robustness and slow down the convergence speed of training loss. Thus, we use RI=1/5 to generate the dataset and then use this dataset to perform the following experiments.

#### 4.2.2. BLSTM Structure Testing

We tested the performance of the BLSTM models with different structures that have various hidden layers and hidden neurons in each hidden layer. This experiment was performed in dataset RI(1/5), and the training labels were the trend signal coefficients shown in Figure 4. The depths 1, 2, 3, and 4 were tested. We compared mean square error training losses and used Adam to train the BLSTM model.

The mean final loss values of each compared BLSTM are listed in Table 5. The mean final loss of each structure is the mean of 10 times training. Note that the BLSTM structures are presented in the format of “L*p*(*q*),” in which *p* indicates the layer number of BLSTM layer, and *q* indicates the number of neurons in BLSTM layer *p*. For example, “L1(128) L2(256)” indicates a two-layer BLSTM with 128 neurons in the first layer BLSTM and 256 neurons in the second layer BLSTM. The test was started with one-layer BLSTM, based on the structure with the best mean final losses, and then one additional BLSTM layer was added. We tested structures that have layers from one to four. The structure with the best mean final loss is shown in bold in Table 5, and the loss curve with the best structure and the lowest final loss in each group are plotted in Table 5, BLSTM “L1(128) L2(192)” achieves the best mean final loss values of 0.0258 and “L1(128) L2(192) L3(128) L4(128)” has the worst loss of 0.8284. The two-layer BLSTM performs better than others. Figure 10 indicates that “L1(128) L2(192)” can achieve a better convergence performance and obtain a lower loss than the others. Therefore, the BLSTM “L1(128) L2(192)” is used to perform the following experiments.

### 4.3. EMD Performance Comparison

We evaluated the performance of the proposed BLSTM-based EMD algorithm, as shown in Figure 4. We compared the proposed two-layer BLSTM-based EMD algorithm, which is trained by the RI(1/5) dataset, to the other EMD methods, with respect to the EMD performance on our MEMS inertial datasets. The compared methods were as follows: traditional EMD [43], multiobjective optimization-based EMD [48], frequency-shifting-based EMD [49] and noise assistance analysis-based EMD [50]. We first evaluated EMD performance that includes the IMFs and Hilbert spectrum of the proposed method and the orthogonality of the IMFs. Then we compared the EMD computing efficiency, which is the time consumption in various data lengths.

#### 4.3.1. Evaluation of EMD Results

We evaluated the EMD performance in terms of decomposition performance on different frequency components. Figure 11 and Figure 12 display the EMD results and the corresponding Hilbert spectrums of the proposed BLSTM-based method. Figure 11 shows that the proposed BLSTM-based EMD can decompose various IMFs of a signal, and the features of each fault can be represented according to the IMFs. Figure 12 shows the Hilbert spectrums of the IMFs. It can be seen that the time–frequency features of each fault differ.

To evaluate the IMF decomposing performance, we compared the orthogonality index criterion presented by Bagherzadeh et al. [48] as follows:(26)$$OI=\frac{\int_{t}IMF_{i}\left(t\right)\cdot \left(r_{i}\left(t\right)-IMF_{i}\left(t\right)\right)dt}{\int_{t}^{}r_{i}\left(t\right)dt}$$
where $r_{i}\left(t\right)$ indicates the residual for computing $IMF_{i}\left(t\right)$.

Table 6 lists the orthogonality index criterion of each fault in the proposed method and the compared methods. For the proposed EMD method, the orthogonality index criterion in each fault has small values, which means that the proposed method can decompose various frequency components orthogonally. The orthogonality index values of the proposed method and the compared methods are approximate values because our dataset for training BLSTM is generated by the algorithm which is improved from current methods.

#### 4.3.2. Comparison of EMD Efficiency

We compared the EMD efficiency between our proposed EMD method and other EMD methods in terms of the time required for performing an EMD operation. To perform this comparison, we chose various inertial measurement unit (IMU) data with different lengths, which were 11,996, 16,679, 20,836, 25,726, 29,899, 33,002, 37,086, 42,429, 46,861, and 50,854. Results are shown in Figure 13, Our proposed method had the lowest time consumption, with a mean of 7.6475 s for each EMD. The EMD method in 45 had the longest time, 23.4739 s, for performing each EMD, whereas the traditional EMD took 12.0634 s to perform each EMD. The EMD methods in 46 and 47 took approximately the same amount of time as the traditional EMD, 13.2865 s and 14.1303 s respectively, to perform each EMD.

As in our proposed method EMD directly operates with BLSTM, our algorithm does not need various inner loop iterations. The largest time consumption of an EMD method in [48] is due to the time required for the genetic algorithm to perform the evolution computing. The similarity in time consumption among the EMD methods of [49,50] and traditional EMD is due to these three methods having the same sifting manner in EMD operation.

### 4.4. Multiscale CNN Comparison

We tested the classification accuracy of the proposed multiscale CNN in the case of different fully connecting (FC) layer configuration. The results are shown in Figure 14. We first tested the one-layer FC performance in a different number of neurons to find the best one-layer FC configuration, then added an FC layer based on the optimal one-layer FC structure and found the two-layer configuration structure with the best classifying performance. The classification accuracy was evaluated according to a 10-times mean accuracy test. The metrics we chose were minimum, maximum, mean, and standard deviation. The adam optimizer and cross-entropy were used to train the CNN. The training epochs were set to 200, and the batch size was 16.

Figure 14 shows the mean classification accuracy for different FC configurations. Note that “FC-i” indicates the one-layer FC configuration and “FC-i-j” indicates the two-layer FC configuration, and that “i” indexes the neurons in the first FC layer and “j” indexes the neurons in the second FC layer. For the one-layer CNN, the FC-512 has the highest mean accuracy of 94.9992%, with a standard deviation of 0.3142. Then we added one FC layer to the network “FC-512.” We found that the network “FC-512-128” obtained the best mean classification accuracy of 97.1571%, with a standard deviation of 0.3218. Therefore, the multiscale CNN, FC-512-128, was adopted for FD operation.

### 4.5. FD Performance Comparison

We compared our proposed FD method with five state-of-the-art data-driven FD methods: Kordestani et al.’s ANN-based method [21], Baskaya et al.’s SVM-based method [31], Guo et al.’s CNN-based method [20], Yang et al.’s HHT-CNN-based method [43] and Wen et al.’s CNN-based method [38]. We set training epochs to 300 and set the batch size to 16. The cross-entropy and the Adam optimizer were used to train the neural networks. We performed the comparisons on mean accuracy and confusion matrix to evaluate the accuracy and the misclassified performance.

#### 4.5.1. Comparison on Mean Accuracy

We compared the mean fault classification accuracy by performing our inertial dataset on each method. The results, as shown in Figure 15, indicate the mean, minimum and maximum classification accuracy after being run 10 times. The results show that our method achieved the best fault classification accuracy, at 97.3007%. The CNN-based methods, which are proposed by Guo et al. [20], Yang et al. [43] and Wen et al. [38] achieved mean FD accuracies of 94.6702%, 95.3258%, and 95.8644%, respectively. The machine learning-based methods, which are Kordestani et al.’s ANN-based method [21] and Baskaya et al.’s SVM-based method [31] achieved FD accuracies of 90.1147% and 91.1357%, respectively.

#### 4.5.2. Comparison on Confusion Matrix

We further compared the misclassification performance by using confusion matrix analysis. Figure 16 shows the FD accuracy confusion matrix of each FD method. Our proposed method obtained the best accuracy, at 97.2434%. The fault 1 and fault 3 achieved lower accuracies, respectively, whereas fault 0 had the worst accuracy, and fault 2 achieved the best accuracy. Kordestani et al. [21] had the worst total accuracy of 90.0904% with the largest misclassification. Similar to the mean accuracy results, the CNN-based methods, which are proposed in 20, 43, and 38, have smaller misclassifying results than the machine learning-based FD methods proposed in [21,31].

The results show that the CNN-based methods performed better than the machine learning-based methods, benefited by the advantages of the DL algorithm in handling complex signals. The HHT- and CNN-based methods, which are the proposed methods and Yang et al.’s method [43], performed better than the time domain sliding window-based method [38] and the shot time Fourier transform-based method [20], proving that HHT has obvious advantages in handling nonstationary and non-linear signals. The best performance was achieved by our proposed FD method, which benefits from the proposed HHT-based feature extraction method and the multiscale CNN. The proposed HHT-based feature extraction method has advantages in mining nonstationary features because the use of HHT and the combination of frequency shifting and EEMD improves the EMD performance by resolving the mode mixing problem. The multiscale CNN combines the shallow features and deep features to feed more detailed information to classify the faults, which increases the FD classification performance.

## 5. Conclusions

In this paper, we propose a method for fault diagnosis of MEMS inertial sensors by combining BLSTM-based HHT and CNN. The inertial feature extraction is performed by our proposed BLSTM-based HHT, which offers high EMD efficiency through the use of a priori knowledge of MEMS inertial data. A training dataset generation algorithm for training the BLSTM model of the proposed BLSTM-based HHT is proposed. The task of fault classification is performed by the proposed MS-CNN. The experiment results show that the proposed BLSTM-based EMD has excellent computation efficiency and achieved satisfactory decomposing performance and that the proposed CNN can precisely classify the fault patterns.

Comparing with the relevant FD methods, the proposed FD method has following strengthens: (1) the HHT-based feature extraction algorithm offers advantages in processing variable-temperature fault features. (2) the directly HHT-based feature extraction algorithm by combining BLSTM and HHT offers the obvious time consumption advantages. (3) CNN offers advantages in fault features mining. Note that the preferable feature coding strategy of BLSTM-based EMD is not considered in our paper, that makes BLSTM-based EMD operation also needs additional time frequency shifting and EEMD to improve its performance. In the future, an improved feature coding method that contains the state of the EEMD and frequency shifting can be studied for integrating the additional operation into the BLSTM-based regression model in order to further improve the efficiency of EMD.

## Figures and Tables

**Figure 1 sensors-20-05633-f001:**
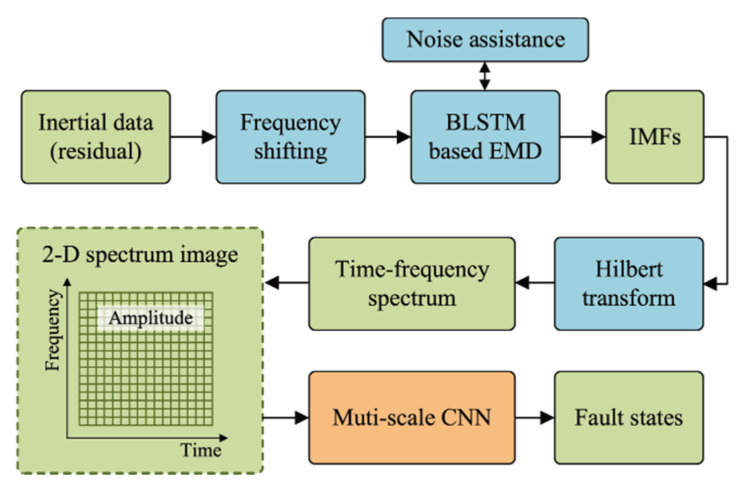
Process of the proposed fault diagnosis method.

**Figure 2 sensors-20-05633-f002:**
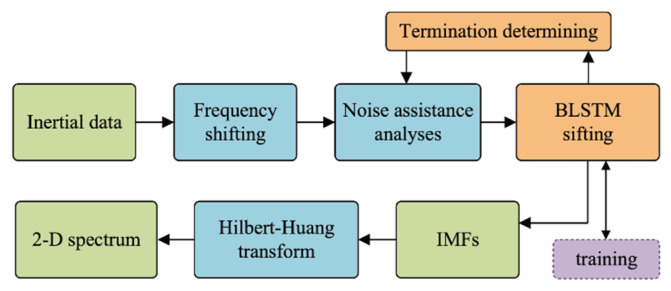
Proposed BLSTM-based HHT structure.

**Figure 3 sensors-20-05633-f003:**
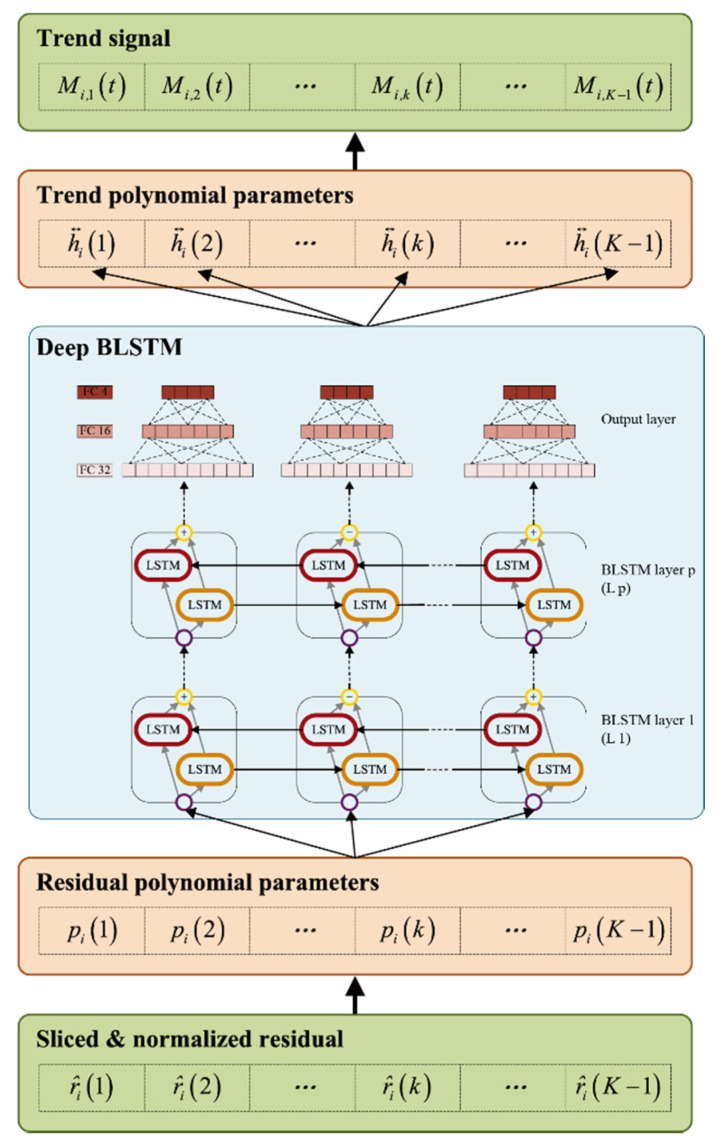
BLSTM-based sifting algorithm.

**Figure 4 sensors-20-05633-f004:**
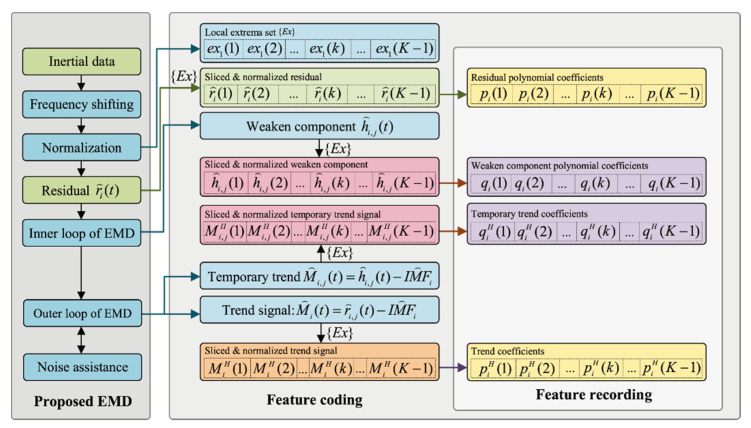
Modified EMD with data recording.

**Figure 5 sensors-20-05633-f005:**
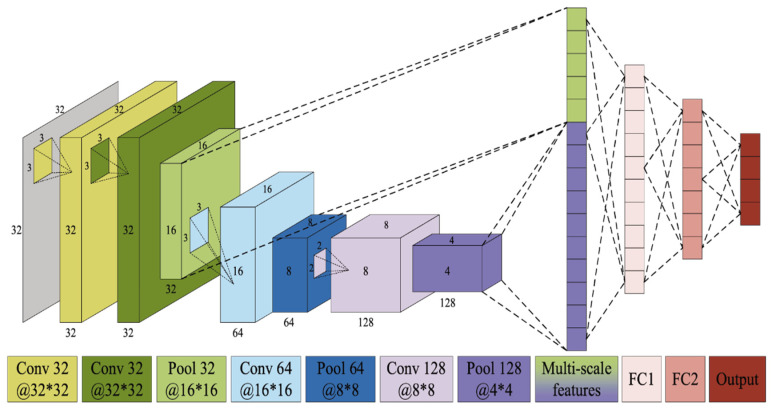
Structure of MS-CNN.

**Figure 6 sensors-20-05633-f006:**
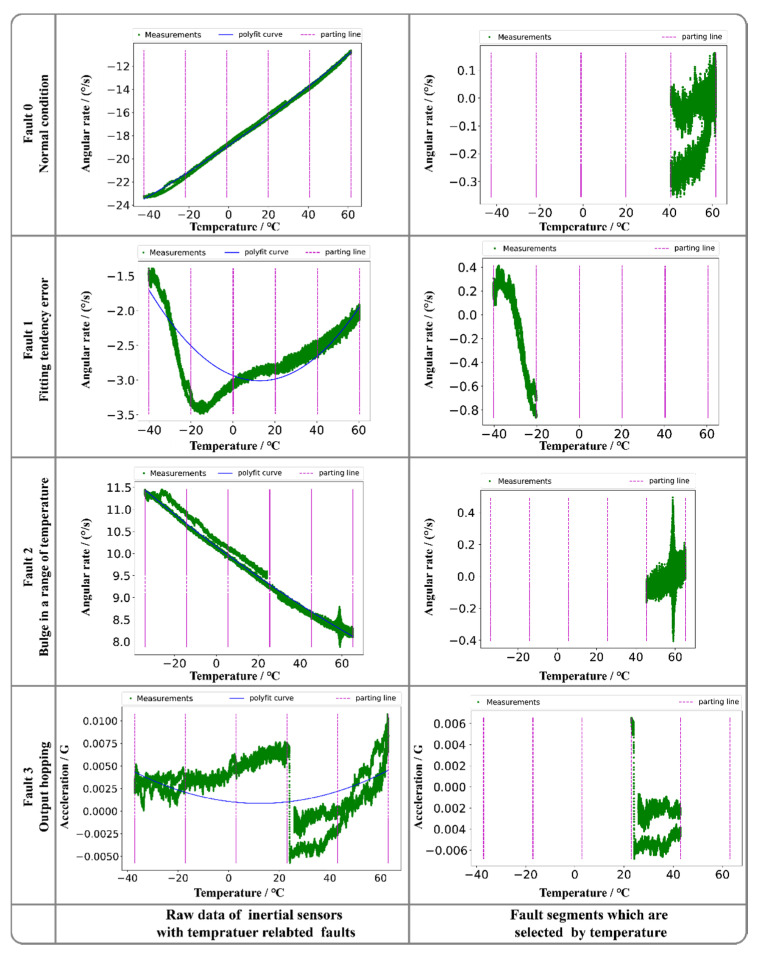
Temperature-related inertial sensors faults.

**Figure 7 sensors-20-05633-f007:**
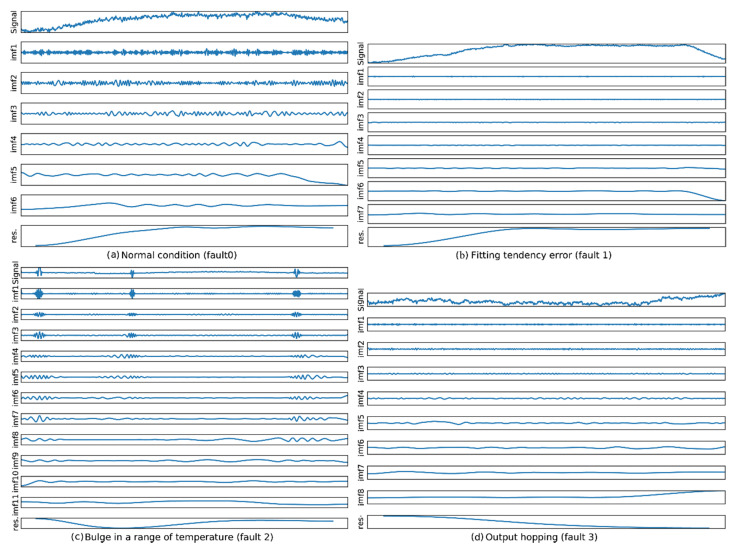
IMFs of each inertial sensor fault. (**a**) the IMFs in case of normal condition; (**b**) the IMFs in case of fault 1 occurs; (**c**) the IMFs in case of fault 2 occurs; (**d**) the IMFs in case of fault 3 occurs.

**Figure 8 sensors-20-05633-f008:**
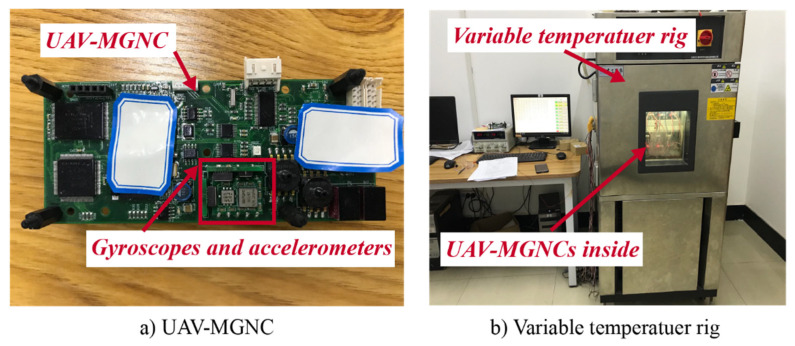
The descriptions of UAV-MGNC and variable temperature rig. (**a**) Description of UAV-MGNC, gyroscopes and accelerometers are the build-in components of UAV-MGNC; (**b**) Description of variable temperature rig, UAV-MGNCs are set inside of variable temperature rig.

**Figure 9 sensors-20-05633-f009:**
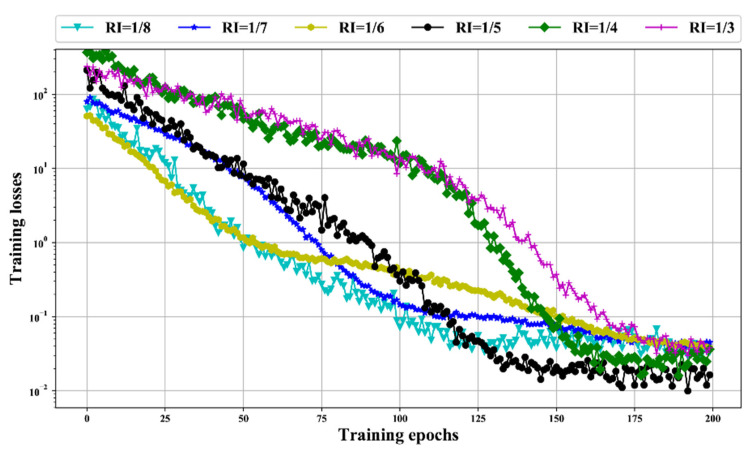
BLSTM training losses in Semi-log coordinate using data sets with various RIs.

**Figure 10 sensors-20-05633-f010:**
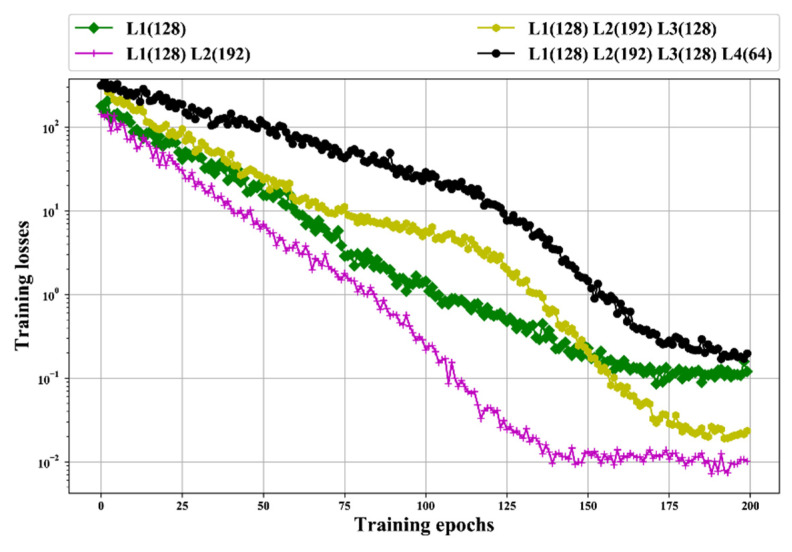
Loss curves in semi-log coordinate with the best structure and the lowest final loss in each group.

**Figure 11 sensors-20-05633-f011:**
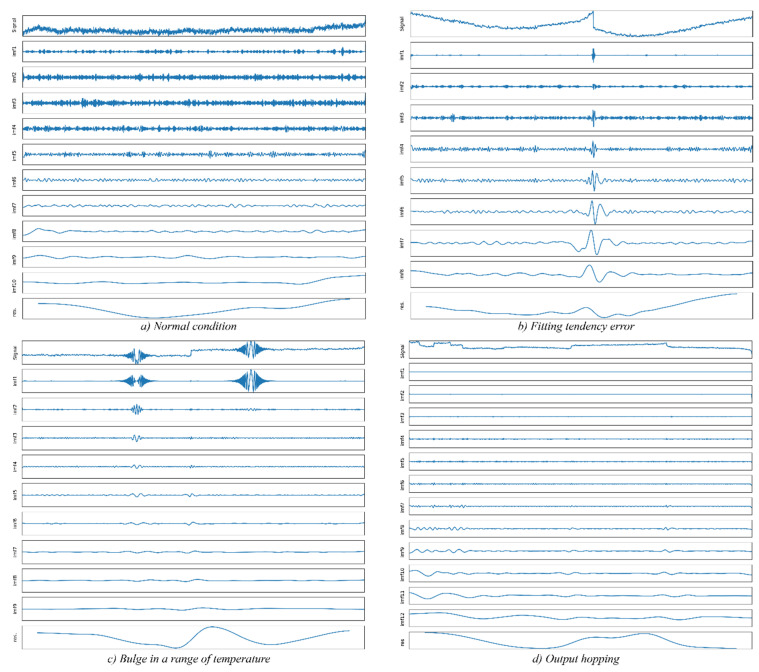
EMD results using proposed BLSTM-based EMD. (**a**) EMD result in case of normal condition; (**b**) EMD result in case of fitting error fault occurs; (**c**) EMD result in case of bulge in a range of temperature fault occurs; (**d**) EMD result in case of output hopping error fault occurs.

**Figure 12 sensors-20-05633-f012:**
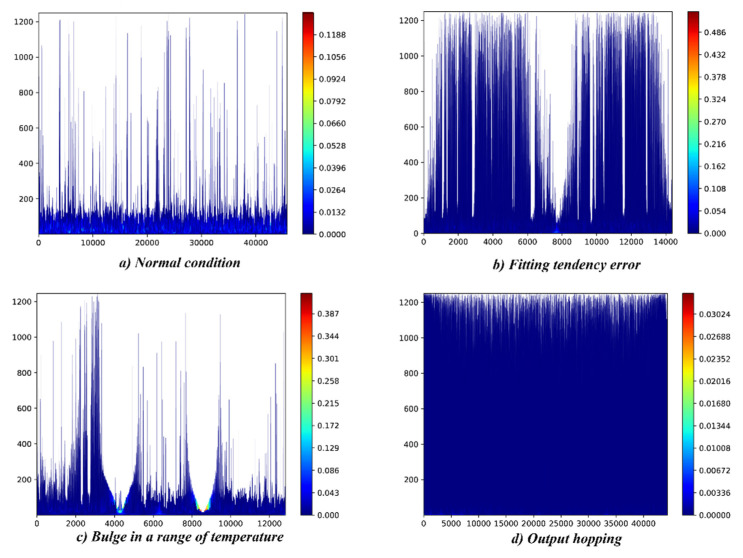
Hilbert spectrum based on EMD results which are shown in Figure 11. (**a**) Hilbert spectrum in case of normal condition; (**b**) Hilbert spectrum in case of fitting error fault occurs; (**c**) Hilbert spectrum in case of bulge in a range of temperature fault occurs; (**d**) Hilbert spectrum in case of output hopping error fault occurs.

**Figure 13 sensors-20-05633-f013:**
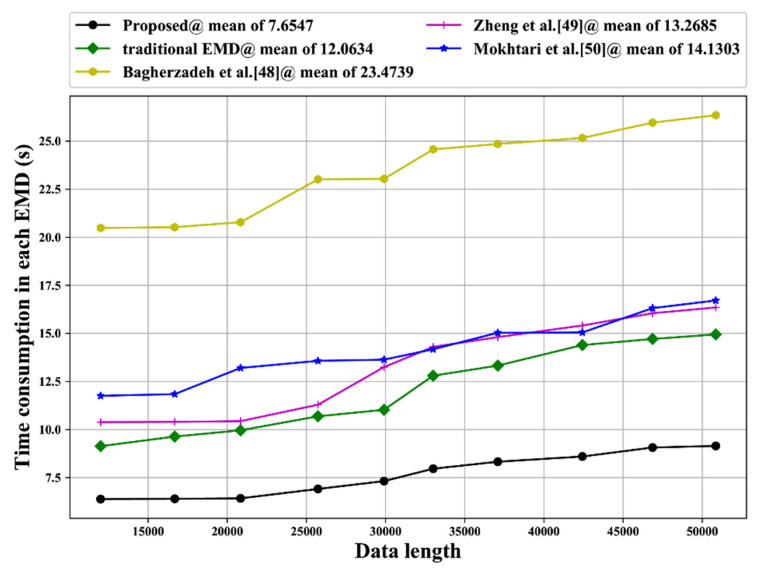
Time consumption comparison.

**Figure 14 sensors-20-05633-f014:**
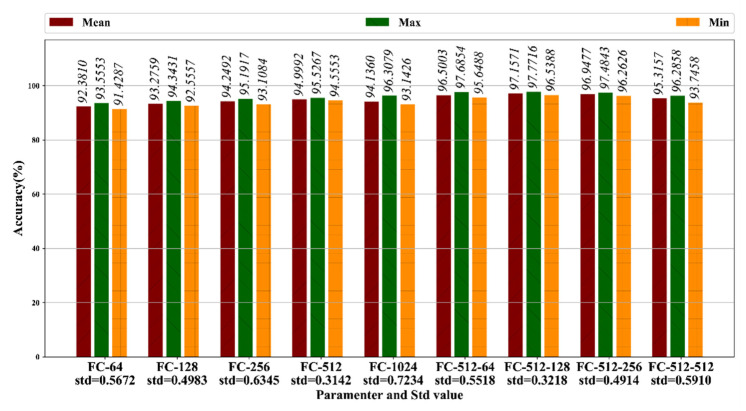
Accuracy test results (in percent).

**Figure 15 sensors-20-05633-f015:**
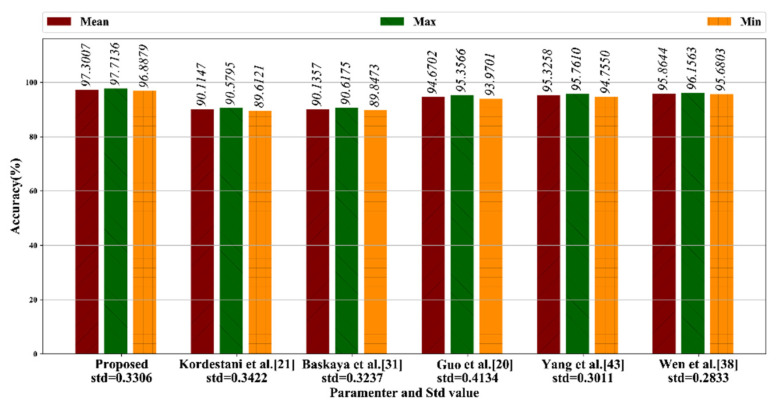
Fault classification accuracy comparison (in percent).

**Figure 16 sensors-20-05633-f016:**
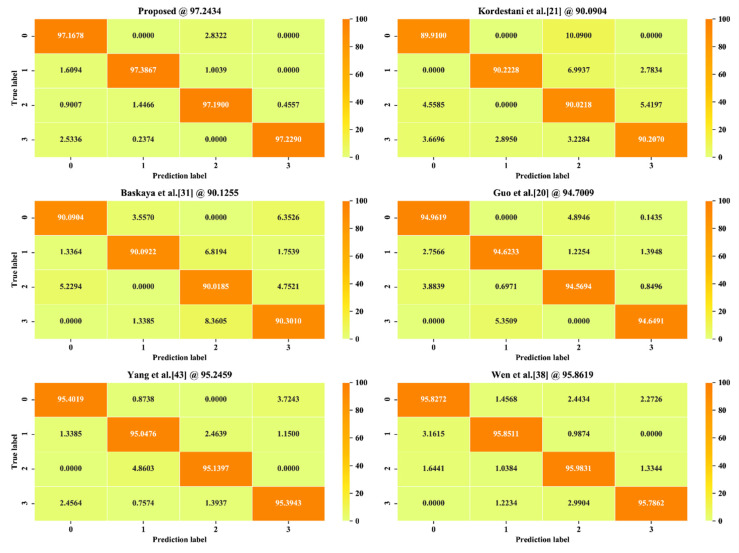
Confusion matrix comparison (in percent).

**Table 1 sensors-20-05633-t001:** Multi-scale CNN configuration.

Layer	Name	Details
1.	Conv1	Conv (3×3×32); stride: (1×1)
2.	Conv2	Conv (3×3×32); stride: (1×1)
3.	Pool 1	Max pool (2×2×32); stride: (1×1)
4.	Conv3	Conv (3×3×64); stride: (1×1)
5.	Pool 2	Max pool (3×3×64); stride: (1×1)
6.	Conv4	Conv (2×2×128); stride: (1×1)
7.	Pool 3	Max pool (2×2×128); stride: (1×1)
8.	Multi	Concatenate features from layer 3 and 7
9.	FC1	Fully connect 1
10.	FC2	Fully connect 2
11.	Output	Soft max (4)

**Table 2 sensors-20-05633-t002:** MEMS inertial sensors fault data set.

State	Size	Label	Label
Normal condition	6000	0	1000
Fitting tendency error	6000	1	0100
Bulge in a range of temperature	6000	2	0010
Output hopping	6000	3	0001

**Table 3 sensors-20-05633-t003:** Selected parameters.

Parameter	Description	Selected Value
Batch size	Training samples in each training epoch	16
Dropout rate	Dropout probability	0.25
Training epochs	Number of training iterations	200
Unfold steps	Data length of each training sample in time steps	256

**Table 4 sensors-20-05633-t004:** Recording process details.

RI	Dataset	Name
1/8	$\left\{{\tilde{h}}_{i,J}\left(k\right),\left(a_{i,J,k}^{H},b_{i,J,k}^{H},c_{i,J,k}^{H},d_{i,J,k}^{H}\right)\right\},J\in \left[2,\mathrm{int}\left(\left(1/8\right)\times \max\left(j\right)\right)\right]$	RI (1/8)
1/7	$\left\{{\tilde{h}}_{i,J}\left(k\right),\left(a_{i,J,k}^{H},b_{i,J,k}^{H},c_{i,J,k}^{H},d_{i,J,k}^{H}\right)\right\},J\in \left[2,\mathrm{int}\left(\left(1/7\right)\times \max\left(j\right)\right)\right]$	RI (1/7)
1/6	$\left\{{\tilde{h}}_{i,J}\left(k\right),\left(a_{i,J,k}^{H},b_{i,J,k}^{H},c_{i,J,k}^{H},d_{i,J,k}^{H}\right)\right\},J\in \left[2,\mathrm{int}\left(\left(1/6\right)\times \max\left(j\right)\right)\right]$	RI (1/6)
1/5	$\left\{{\tilde{h}}_{i,J}\left(k\right),\left(a_{i,J,k}^{H},b_{i,J,k}^{H},c_{i,J,k}^{H},d_{i,J,k}^{H}\right)\right\},J\in \left[2,\mathrm{int}\left(\left(1/5\right)\times \max\left(j\right)\right)\right]$	RI (1/5)
1/4	$\left\{{\tilde{h}}_{i,J}\left(k\right),\left(a_{i,J,k}^{H},b_{i,J,k}^{H},c_{i,J,k}^{H},d_{i,J,k}^{H}\right)\right\},J\in \left[2,\mathrm{int}\left(\left(1/4\right)\times \max\left(j\right)\right)\right]$	RI (1/4)
1/3	$\left\{{\tilde{h}}_{i,J}\left(k\right),\left(a_{i,J,k}^{H},b_{i,J,k}^{H},c_{i,J,k}^{H},d_{i,J,k}^{H}\right)\right\},J\in \left[2,\mathrm{int}\left(\left(1/3\right)\times \max\left(j\right)\right)\right]$	RI (1/3)

**Table 5 sensors-20-05633-t005:** Mean final losses of each BLSTM structure.

Group	BLSTM Structure	Mean Loss
1 layer	L1(64)	0.5198
	L1(128)	0.2109
	L1(192)	0.3917
	L1(265)	0.2392
2 layers	L1(128) L2(64)	0.0317
	L1(128) L2(128)	0.0301
	L1(128) L2(192)	0.0258
	L1(128) L2(256)	0.0279
3 layers	L1(128) L2(192) L3(64)	0.0882
	L1(128) L2(192) L3(128)	0.0791
	L1(128) L2(192) L3(192)	0.1003
	L1(128) L2(192) L3(256)	0.0932
4 layers	L1(128) L2(192) L3(128) L4(64)	0.4503
	L1(128) L2(192) L3(128) L4(128)	0.8284
	L1(128) L2(192) L3(128) L4(192)	0.5017
	L1(128) L2(192) L3(128) L4(256)	0.6697

**Table 6 sensors-20-05633-t006:** Orthogonality index criterion.

Method	Fault0	Fault1	Fault2	Fault3
Proposed EMD	$5.0601\times 10^{-5}$	$1.0698\times 10^{-4}$	$2.6451\times 10^{-5}$	$2.8746\times 10^{-4}$
Traditional EMD	$4.8373\times 10^{-4}$	$6.8474\times 10^{-3}$	$4.8372\times 10^{-4}$	$3.3746\times 10^{-5}$
Bagherzadeh et al. [48]	$8.3865\times 10^{-5}$	$3.9473\times 10^{-4}$	$8.4937\times 10^{-4}$	$1.3836\times 10^{-3}$
Zheng et al. [49]	$4.8362\times 10^{-5}$	$7.8924\times 10^{-5}$	$9.3837\times 10^{-4}$	$8.3972\times 10^{-4}$
Mokhtari et al. [50]	$4.9732\times 10^{-4}$	$5.1198\times 10^{-4}$	$4.9471\times 10^{-5}$	$3.7765\times 10^{-4}$

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
