# Peer review of "Method for Fault Diagnosis of Temperature-Related MEMS Inertial Sensors by Combining Hilbert–Huang Transform and Deep Learning"

_sensors, 2020, doi:10.3390/s20195633_

Round 1

Reviewer 1 Report

Line 81 to 95: It’s a strange art to present contents using such bullets in the paper.

Line 355: Descriptions in the figures are too small to read. Generally the figure captions are not enough in detail to state the figures clear.

Generally the background knowledge is very extensively stated, while experiments are also noted in details. For the readers it would be helpful to have some comparisons with relevant fault diagnosis methods for a lateral comparison and stating the strengthes and shortcomings of this work.

Author Response

We sincerely thank the anonymous reviewers for their detailed, insightful and valuable comments. We have carefully considered the reviewers’ comments and revised the paper accordingly. Please find attached revisions.

Reviewer 2 Report

The paper proposes a method for fault diagnostic of MEMS) inertial sensors using bidirectional long short-term memory (BLSTM)-based Hilbert–Huang transform (HHT) and convolutional neural network (CNN). The paper is well-written. A have just a few comments for the authors. 1) The combination of HHT and CNN is recurrently employed in the literature. There are many applications in different fields as follow. https://ieeexplore.ieee.org/document/8698326 https://journals.sagepub.com/doi/full/10.1177/1550147719888169 Hence, I miss a literature review about this combination. The authors focused more on separated topics about HHT and CNN. I suggest the authors to add some papers which focus on the combination. 2) Several methods have been proposed in the field of SHM the usage of CNN as a classification tool. In the section 2.1 the authors could go further. A critical review of the literature could be addressed. A quick search in the Sensor’s homepage we can find many papers. For example, two of them are listed below. https://www.mdpi.com/1424-8220/19/22/4933 https://www.mdpi.com/1424-8220/18/9/2955

Author Response

(The authors gave the same response as above.)

Reviewer 3 Report

  1. LINE 63 has a grammatical error.
  2. There seems to be too many significant figures on line 512~517 to be redundant.
  3. Since the focus of this article is on temperature-related faults of MEMS inertial sensor, the title of "UAV "becomes redundant. At the same time, this analysis is limited to temperature effects, and it is more appropriate to use "...Fault Diagnosis of temperature-related MEMS inertial sensor.
  4. There are few descriptions about the experimental settings and conditions. It is best to add more descriptions about it.

Author Response

(The authors gave the same response as above.)
